# Polyandry blocks gene drive in a wild house mouse population

Andri Manser [1,2✉], Barbara König [1] & Anna K. Lindholm [1]

Gene drives are genetic elements that manipulate Mendelian inheritance ratios in their favour. Understanding the forces that explain drive frequency in natural populations is a long-standing focus of evolutionary research. Recently, the possibility to create artificial drive constructs to modify pest populations has exacerbated our need to understand how drive spreads in natural populations. Here, we study the impact of polyandry on a well-known gene drive, called *t* haplotype, in an intensively monitored population of wild house mice. First, we show that house mice are highly polyandrous: 47% of 682 litters were sired by more than one male. Second, we find that drive-carrying males are particularly compromised in sperm competition, resulting in reduced reproductive success. As a result, drive frequency decreased during the 4.5 year observation period. Overall, we provide the first direct evidence that the spread of a gene drive is hampered by reproductive behaviour in a natural population.

[1] Department of Evolutionary Biology and Environmental Studies, University of Zurich, Winterthurerstrasse 190, Zurich, Switzerland. [2] Department of Evolution, Ecology and Behaviour, University of Liverpool, Biosciences Building, Crown Street, Liverpool, UK. ✉email: Andri.Manser@ieu.uzh.ch

The fair 50% propagation of alleles under Mendelian inheritance is a fundamental feature of diploid life. Gene drives are stretches of DNA that disobey Mendel's rules. By manipulating gametogenesis or meiosis of diploid organisms in their favour[1], they ensure transmission to systematically more than 50% of progeny. This deviation from Mendelian expectation is termed gene drive. Gene drive allows drive elements to rapidly spread in populations even if they incur substantial fitness costs to the organisms and populations that harbour them[2].

Recently, the potential for rapid spread in natural populations in spite of organismal fitness costs has sparked enormous interest in gene drive as a tool in pest control (as genes useful for human ends are often costly to the carrier organism). The release of artificial drive constructs into target populations could be used to eradicate or genetically modify pest species, e.g. by spreading a synthetic drive construct that inhibits the transmission of insect-born pathogens. Technological advances in genome editing make the genetic engineering of artificial drive constructs at an affordable price a real possibility and could revolutionize the way we deal with diseases and invasive species[3–5]. However, the emerging technology also confronts us with enormous challenges. For example, and in contrast to rapid developments in molecular tools to construct synthetic drivers, we still know very little about what will happen if such constructs are released into natural populations. In particular, we still have a poor understanding of the factors that determine the frequency of gene drive in wild populations. Naturally occurring drive systems present us with the great opportunity to study the evolutionary dynamics of drive in a biologically relevant setting. Yet despite over half a century of extensive research effort, our ability to predict the fate of drive elements in natural populations remains extremely limited[2].

A factor that has been identified to play a key role for the spread of drive, both theoretically[6–8] and under laboratory conditions[9–12], is polyandry (the mating of a female with more than one male in a single reproductive event) and subsequent sperm competition[10]. Many known natural drive systems manipulate spermatogenesis in males, which typically involves the killing of sperm that do not carry the driver. While this selective targeting of non-carrier sperm gives the gene drive advantage against the rival chromosome within a male (causing the drive effect), empirical data collected over the past decade has demonstrated that such sperm killing compromises the sperm competitiveness of drive males in competition against other males. Laboratory studies on a range of taxa have shown that drive males suffer from drastically reduced numbers of functional sperm, translating into low fertilization success when in sperm competition against rival wildtype males[9]. The link between gene drive and sperm competition is interesting for at least two reasons. Firstly, we would expect the spread of a gene drive to be limited in species or populations where polyandry is prevalent[6,7,12]. Secondly, gene drive may create selection for increased polyandry rates in females as repeated mating will reduce the probability of ova being fertilised by drive carrying sperm[9]. There is convincing evidence for both effects from laboratory studies[11–14]. However, we currently do not know whether these sperm competitive effects are relevant for gene drive systems in a natural context.

To fill this gap, we here examined the role of gene drive on sperm competitive ability in a natural population of house mice (*Mus musculus domesticus*) that harboured a gene drive system called *t* haplotype. The *t* haplotype is a variant of mouse chromosome 17 carrying several genetic factors that selectively disrupt flagellar function of wildtype + sperm in heterozygous males during spermatogenesis[15]. As a result, +/*t* heterozygous males transmit the *t* to about 90% of the offspring instead of the 50% expected under Mendelian inheritance[16]. Females transmit *t* haplotypes in a Mendelian fashion. As is the case for most naturally occurring drive systems, the *t* haplotype has detrimental fitness consequences for individual carriers. Due to recessive lethal mutations, *t*/*t* homozygote individuals perish *in utero*. The ramifications for a population harbouring the *t* haplotype are dramatic: as a direct result of recessive lethality and drive, +/*t* females suffer from a 40% litter size reduction in monogamous matings with a +/*t* male[16], corresponding to a 10% death rate in the population as a whole (in a randomly mating population). Recent laboratory experiments have further shown that +/*t* males perform poorly in sperm competition, only fertilising about 11% of the offspring when competing with a +/+ male[11]. Interestingly, the magnitude of this effect implies that *t* haplotypes cause even more damage to +/*t* ejaculates than expected by the numeric reduction in sperm number due to + sperm disruption alone[11]. A number of studies have further demonstrated that house mice are actively polyandrous both under laboratory and natural conditions[17–20].

Here, we test four key aspects of the relationship between gene drive and polyandry in a natural population of house mice that has been extensively monitored over a period of 4.5 years. First, we measure the frequency of polyandry in the population by performing a comprehensive analysis of the genetic and environmental factors that affect the occurrence of multiply sired litters using an animal model framework based on a near complete population pedigree. Second, we examine the repercussions of polyandry and sperm competition on reproduction in males. For the first time in a wild population, we show that polyandry reduces the reproductive output of drive-carrying males. Third, we examine whether the poor sperm competitive performance of drive males can explain the low drive frequencies in the population, by comparing them to both monandry and polyandry theoretical predictions. Fourth, we ask whether the presence of the gene drive triggered an evolutionary response for increased polyandry rates in females by quantifying heritability and selection of polyandry.

## Results

**Measuring polyandry.** Among 682 litters from 225 females born between January 2006 and July 2010, 323 were sired by more than one father, corresponding to a polyandry rate of 47.3% (95% CI: [43.2, 51.1%]). The number of sires per litter varied between one and four (Fig. 1a). Note that this polyandry estimate based on paternity outcomes (henceforth referred to as *genetic* polyandry) will likely underestimate the actual *behavioural* polyandry rates, as not every mating results in a successful fertilisation. Based on a mathematical model, we estimate that 13.6% litters were misclassified as monogamous, with one male siring all offspring of a litter despite the female mating with multiple males (see Supplementary Note 4 and Supplementary Table 3 for details). Our best estimate for the behavioural polyandry rate is thus 60.9% (95% CI: [56.2, 65.8%]). A generalised animal model found that the probability of genetic polyandry increased with adult population density (Fig. 1b, posterior slope estimate: 0.87, 95% CI: [0.33, 1.41], $P < 0.01$). As expected, owing to the larger detection probability of genetic polyandry in larger litters, it also detected a positive relationship between litter size and polyandry (posterior slope estimate: 0.26, 95% CI: [0.14, 0.37], $P < 0.001$). Both female identity and additive genetic variation explained little to no variation in polyandry (Supplementary Fig. 1). As a consequence, the heritability of genetic polyandry was very low, with a posterior mean and 95% CI of $h^2 = 4.32 \times 10^{-4}$ [0, 0.12] (Supplementary Note and Fig. 1). Average monthly temperature and female *t* genotype had no statistically relevant effect on polyandry rates and were removed from the model.

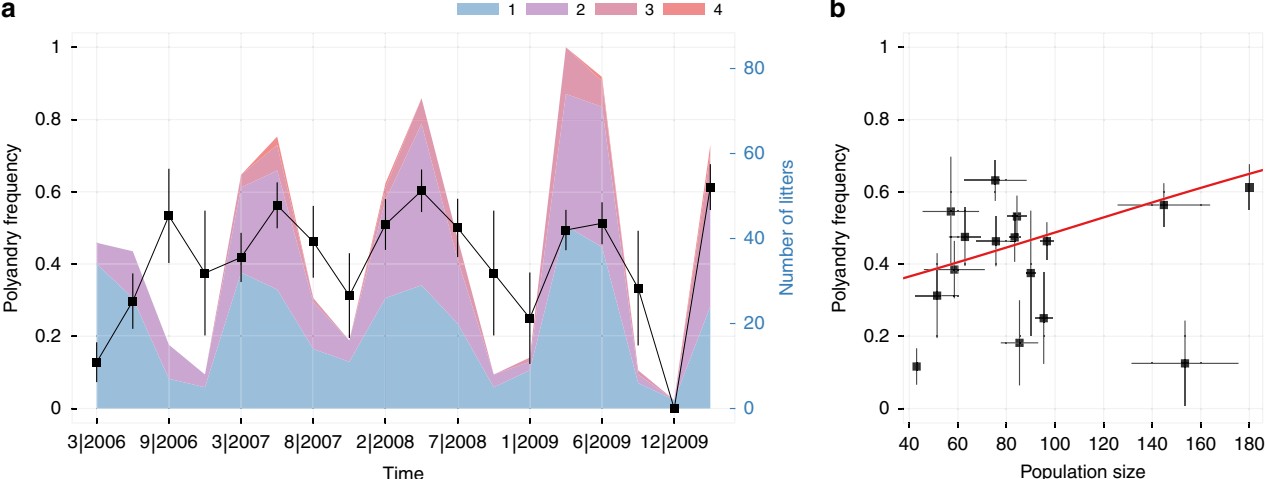

**Fig. 1 The frequency of polyandry in the study population. a** Black line and squares depict the frequency of genetic polyandry among the 682 litters (proportion of litters with >1 father, mean ± binomial standard error) during the observation period. Absolute numbers of monogamous litters (sired by one male only) and polyandrous litters are shown in blue and violet to red gradient colours (representing two, three, and four sires), respectively. **b** Positive relationship between the frequency of polyandry and adult population density as predicted by the generalized animal model (red line). The raw data were grouped in quarter-year time intervals, indicating mean (squares) and binomial standard errors (vertical lines) in genetic polyandry rates, as well as mean (squares) and standard errors (horizontal lines) in adult population size. Source data are provided as a Source data file.

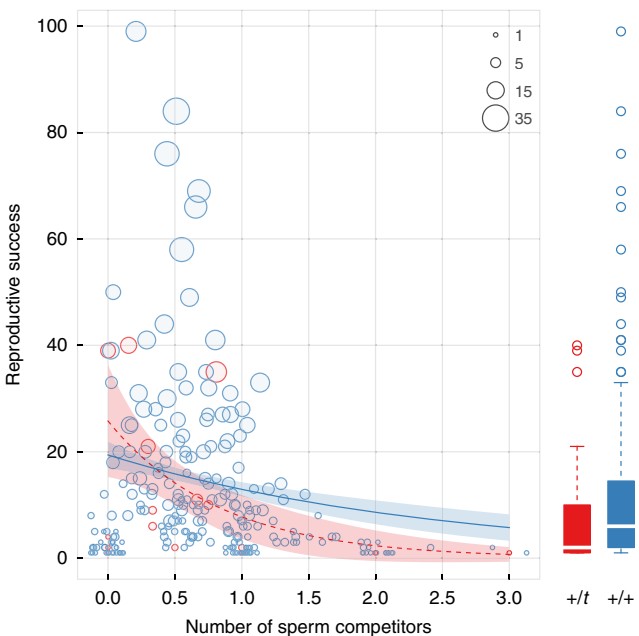

**Fig. 2 Sperm competition damages the fitness of *t* haplotype males.** The figure shows the reproductive success of 249 males as a function of sperm competition intensity (measured as the number of sperm competitors a male encountered during his lifetime) and *t* genotype (red for +/*t* and blue +/+ males). Dot size is proportional to the number of litters in which a male sired at least one offspring. Red/dotted and blue/solid lines with shaded areas show mean GLM model predictions and 95% confidence bands for +/*t* and +/+ males, respectively (based on a hypothetical male that reproduced 10 times). Reproductive success was lower in males that were exposed to elevated levels of sperm competition, but the effect is particularly strong in drive carrying males. Low drive male fitness under sperm competition translated into a lower reproductive output of +/*t* males overall, as illustrated by boxplots on the right (showing median (line), 0.25 and 0.75 quartiles (box) ± 1.58 times the inter-quartile range (whiskers)). Source data are provided as a Source data file.

**The impact of polyandry on male fitness.** 249 males successfully sired at least one offspring during the observation period. Average reproductive success among them was 11.32 offspring (range: [1, 99]) distributed over 4.8 litters (range: [1, 35], Fig. 2). Males had to compete against 0.76 or 0.56 other males based on arithmetic and harmonic mean, respectively, ranging from 0 to 3. According to the best generalised linear model (Fig. 2 and Supplementary Table 3), +/+ and +/*t* males did not differ in their reproductive success when mating monogamously (difference in intercepts between +/*t* and +/+ males: 0.62, SEM = 0.54, $P = 0.25$). Males that had to compete against more sperm competitors saw a decrease in their reproductive success (slope estimate for +/+ males: −1.20, SEM = 0.37, $P < 0.01$). Importantly, we found that the effect of sperm competition was significantly stronger in +/*t* males, as indicated by a significant interaction between sperm competition and *t* genotype (difference in sperm-intensity slopes between +/*t* and +/+ males: 0.79, SEM = 0.38, $P < 0.05$). Finally, as expected, our best model showed that reproductive success increased if a male genetically contributed to more litters (slope estimate: 0.14, SEM = 0.02, $P < 0.001$). The intensity of sperm competition, the *t* genotype, and their interaction explained 72.3, 6.6 and 11.4% of the differences (deviance) in male fertilisation success (once accounted for number of litters).

**The impact of polyandry on drive frequency dynamics.** The population was established in 2002 with 12 randomly selected animals from neighbouring farms, 4 of which were *t* carriers, and *t* frequencies have been on a steady decline ever since[7]. During the observation period of this study, the average frequency of drive carriers among the 3,126 newborns was 11.8% (Fig. 3a). This observation is substantially lower than expected by the monandry model, which predicts 66% +/*t* heterozygotes among newborns (Fig. 3 and Supplementary Note 3A). Overall, the polyandry model (Supplementary Note 3A) shows that sperm competition can substantially reduce or even eradicate *t* frequency, depending on the exact levels of polyandry and +/*t* male sperm competitiveness (Fig. 3b). From controlled laboratory experiments, we know that +/*t* males only sire 11.3% (95% CI:

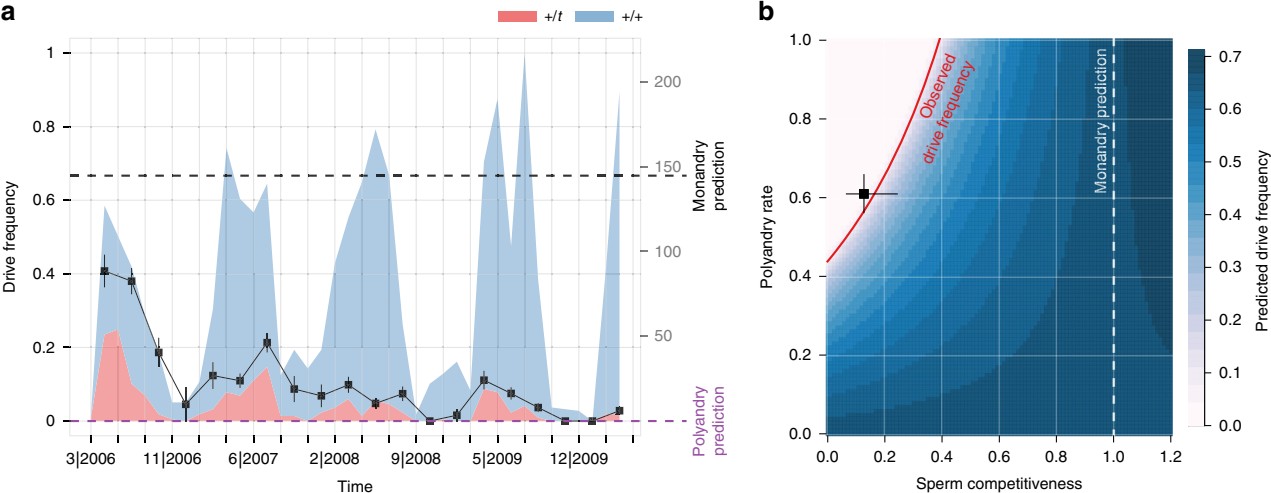

**Fig. 3 The impact of polyandry on drive frequency dynamics.** Observed *t* frequencies in the study population were lower than expected by monandry, but are in line with polyandry model predictions. **a** The frequency of +/*t* heterozygotes among 3126 offspring over the 4.5 year observation period in relative (black squares and lines for mean and binomial standard error, respectively) and absolute numbers (blue/red shaded areas). The grey dotted line shows the expected 66% +/*t* genotypes expected under monandry. The violet dashed line denotes the 0% prediction based on observed levels of polyandry and +/*t* male sperm competitive disadvantage. The polyandry model correctly predicts the observed frequency trend. **b** Predicted and observed +/*t* genotype frequency as a function of +/*t* male sperm competitiveness and polyandry rate (for drive strength $d = 0.9$) based on the polyandry model (Supplementary Note 3). The dotted line denotes the monandry prediction of 66% (where +/*t* males have no sperm competition disadvantage). The red curve corresponds to the average observed drive genotype frequency during the observation period (11.8%). The square with horizontal and vertical lines shows the empirically measured mean and 95% confidence intervals for both parameters (Supplementary Note 4, polyandry rate: 682 litters, sperm competitiveness[11]: 57 mating trials). Again, observed genotype frequencies agree with the (parameterised) polyandry model, but do not align with the monandry prediction. Source data of observed +/*t* genotype frequency are provided as a Source data file.

[6.2, 19.6%]) of offspring when competing against wildtype males[11]. If we insert this estimate into the model (Fig. 3 and Supplementary Note 4 and Supplementary Table 2), we find that *t* will be eradicated whenever polyandry rates exceed 58% (95% CI: [52, 74%]). With an estimated polyandry rate of 60.9% (see above), the fully parameterised polyandry model thus predicted the longterm *t* extinction for this particular population, which is in agreement with the observed frequency trend (Fig. 3a).

**The impact of polyandry on female fitness**. We quantified the impact of polyandry levels per female (as estimated by our animal model) on her reproductive output. The model did not find evidence for directional selection on polyandry, nor did we detect systematic differences in selection gradient (effect of polyandry on fitness) between the *t* genotypes (+/+ or +/*t*, Supplementary Note and Fig. 2, Supplementary Table 1). Moreover, as mentioned above, the generalised animal model detected little to no additive genetic variation for polyandry, suggesting that the heritability of polyandry may be negligible in the measured context (Supplementary Note 1).

**Discussion**
We have elucidated key aspects of the relationship between polyandry and gene drive in a natural population of house mice. First, we established that polyandry was common in our wild house mouse population, with an estimated polyandry frequency of 61% during the observed 4.5 year study period. This had important ramifications for post-mating selection in males—drive carrying males were heavily compromised by sperm competition, translating into a decreased lifetime reproductive output. We go on to show that this sperm competition effect is likely to account for the observed low drive frequencies in the study population. Surprisingly, we find no evidence that the sperm competition effect resulted in the evolution of increased polyandry in females. Overall, we provide the first direct evidence that polyandry suppresses gene drive under natural conditions in any drive system.

The (genetic) polyandry rate of 47% measured here is in broad agreement with previous estimates from both laboratory and natural house mouse populations of 30%–40%[21–23] and 4%–47%[17,18], respectively. Bronson[24] has speculated that higher population densities may allow dominant males to exercise greater control over subordinate males, thereby reducing the opportunity for females to mate multiply. The results provided here and by Dean et al.[17] point to the opposite, suggesting that populations with higher densities increase mating opportunities, resulting in elevated polyandry rates. The fact that polyandry levels were at the upper end of the reported spectrum may reflect the relatively high population density ($2.5 \pm 1.0$ mice m$^{-2}$), but note that they are within the reported range for house mice (up to 10 m$^{-2}$,[24]). Our animal model found little evidence for systematic individual or additive genetic variation of polyandry. Few studies have attempted to quantify heritability of polyandry in a handful of rodent and bird species, and all have reported low heritability estimates, particularly when measured in a natural context[25–27]. This could indicate that polyandry is largely dependent on environmental conditions such as mate availability. Alternatively, the absence of a genetic signal could reflect the limited statistical power. Polyandry is difficult to quantify reliably, particularly when based on paternity, as females only have a handful of litters throughout their life. Moreover, while the paternity approach is convenient as it does not rely on direct behavioural observation, which can be cumbersome in species as elusive as house mice, the method will neglect copulations that do not result in paternity. Such misidentification is particularly pertinent in species with small litters (large sample error) and skewed fertilisation probabilities among males. We here attempted to theoretically infer the proportion of unsuccessful mating attempts, and thus the difference between behavioural and genetic polyandry, based on known drive frequency, sperm competitive disadvantage[11], litter size, and assuming random mating (Supplementary Note 4). The resulting figure of 61% may still underestimate the actual

behavioural polyandry rates, as our model did not account for additional sources of variation in sperm competition (known and unknown), such as mating order, timing of the mating, or copulatory plug formation[11,28,29].

The core finding of the study concerns the repercussions of promiscuity on postmating selection in males. Males who were exposed to elevated levels of sperm competition generally suffered from reductions in their reproductive output. Importantly, this effect was particularly strong for drive males—while drive males and wildtype males had comparable reproductive success in the absence of sperm competition, the reproductive output of drive males was hampered disproportionately with increasing levels of sperm competition. Because the intensity of sperm competition in the population was high overall (due to polyandry), the sperm competitive disadvantage translated into a significant reduction in lifetime reproductive success of drive males relative to wildtype males. This result corroborates laboratory data from house mice[11,12] and a number of insect taxa[14,30–34], which demonstrated that sperm-killing drive systems substantially damage both male fertility and competitive ability when exposed to sperm competition against rival males. Sperm killing drive systems appear to have an important structural weakness that may be an inevitable consequence of their selfish action—by killing or incapacitating rival sperm, a necessary condition to outcompete the rival chromosome *within* the organism, sperm-killing drivers become vulnerable to sperm competition against other males[35]. For the first time, we here show that this structural weakness of gamete killers has relevant fitness implications in a natural context. The sperm effect may have fascinating evolutionary knock-on effects. Runge and Lindholm[36] have hypothesised that drive carriers could evade sperm competition in dense populations via dispersal, and found indeed evidence for higher *t* emigration in the population studied here. Meade et al.[37] show that male stalk-eyed flies adaptively alleviate drive related sperm damage through increased ejaculate investment. For reasons currently unknown, such ejaculate compensation does not seem to occur in house mice[28].

We also expected the low sperm competitiveness of drive males to impact on the evolution of female remating behaviour and, as a result, the species' mating system[10,38]. The selection pressure on females, +/*t* drive carriers in particular, to avoid drive fertilisation is substantial: +/*t* females lose ~40% of their litter when mating monogamously with a +/*t* male due to *t*/*t* embryo lethality[11,16]. By inviting sperm competition via polyandry, a female can minimise the probability of drive fertilisation[11], thus protecting her progeny from *t* lethal effects (analogous to the good-sperm hypothesis[39]). Surprisingly, particularly considering the magnitude of this effect, we did not detect signs of directional selection on polyandry, nor did we find elevated levels of polyandry or selection on polyandry in +/*t* females (note that *t* related costs differ between +/*t* and +/+ females owing to the non-additive/recessive nature of the lethal effect). This could be due to limited statistical power–-despite analysing 682 litters, the expected number of matings among +/*t* heterozygotes is of the order $y^2$ (where $y$ denotes +/*t* frequency), thus as little as ~1% in our case. Offspring number was further measured at the time of genetic sampling (when pups were 13 days old), thus female fitness estimates were conflated with pup survival. Previous work has shown that pup mortality rates in our study population are considerable[20], and disproportionately higher in larger litters[40]. A litter size-dependent mortality could weaken selection for litter size at birth, and thus selection pressure for females to avoid *t* fertilisation.

Understanding the factors that determine the frequency of gene drive in natural populations is a longstanding problem in evolutionary research. Empirically observed frequencies are typically lower than predicted, which suggests the presence of evolutionary forces that limit drive spread[2]. In the context of house mice, the discrepancy between high frequency prediction and low observed frequency[41] is a conundrum that goes back over half a century as the low *t* frequency paradox[42]. Based on theoretical[7] and laboratory work[11,12], we have previously hypothesised that polyandry may explain the discrepancy between empirical observation and theoretical predictions. Solid estimates of polyandry rate (this study) and *t* male sperm disadvantage[11] have finally allowed us to compare the observed *t* frequency dynamics in our study population against a fully parameterised polyandry model. The model predictions suggest that polyandry could go a long way to explain the low observed *t* in natural populations, thus resolving the *t* paradox. Further support for polyandry as a key force for natural *t* frequency dynamics is the observation that *t* frequencies are lower in populations with larger density[41], which is precisely what we would expect if polyandry rates are elevated in high density populations (as shown here). Also note that the impact of polyandry on drive frequency is non-linear as +/*t* males are more likely to encounter (superior) +/+ males in sperm competition when *t* frequencies are low. The *t* frequency dependence of the polyandry effect results in a sudden phase transition from a parameter regime that allows the driver to stably persist to values that cause drive extinction (Fig. 3b). Based on the lab estimate of +/*t* male sperm competitiveness[11], this transition occurs at a polyandry rate of around 50%–70%. Intriguingly, observed polyandry measures from house mice typically fall just below, but relatively close to this transition. Hence, relatively small fluctuations in polyandry and/or +/*t* male competitiveness will translate to marked differences in *t* frequency, a pattern that is again consistent with observations that report marked *t* frequency variation across spatial and temporal scales[41].

While our work highlights the importance of polyandry for low *t* frequencies, it does not rule out the influence of additional factors on *t* dynamics. However, a number of the previously discussed evolutionary forces[42] appear unlikely in the population studied here. Viability selection against +/*t* heterozygotes could hamper *t* frequencies[43], but we did not find survival differences between +/+ and +/*t* males. In fact, +/*t* females even outlived their +/+ wildtype counterparts[7]. Inbreeding will reduce +/*t* heterozygote frequency via a general increase in homozygosity[44,45], yet levels of inbreeding have remained constant in our population[7]. Population substructure and drift within the population appear too small to have tangible effects on *t* dynamics[46,47], but elevated +/*t* migration away from the population may have contributed to the observed *t* frequency drop[36]. Finally, a series of laboratory experiments has found that females avoid +/*t* males based on olfactory cues[48]. Lindholm et al.[16] have analysed the likelihood of paternity among all males in our study population for 2004–2005 (including males that failed to sire any offspring, unlike here), and found that +/*t* males were less likely to sire offspring compared to +/+ males. However, this finding is also compatible with the polyandry effect reported here, as +/*t* males will fertilise fewer eggs even if mating is random, and the lack of information on mating success makes it hard to distinguish between pre- and postmating effects (Supplementary Note 4). We have since measured mate choice under various laboratory settings where mating behaviour could be observed directly, but found no evidence for female discrimination based on male *t* genotype, nor for a difference in female remating probability according to male genotype[49,50]. The fact that +/+ and +/*t* male fitness does not differ in the absence of sperm competition (no intercept difference in Fig. 2) is further evidence for the absence of mate choice in our study population. In general, there are surprisingly few reports of mate choice

against gene drive carriers[10], potentially because its evolution requires a signal that reliably indicates the presence of the driver[51].

The possibility to modify pest populations via synthetic gene drive exacerbates our need to understand the forces that determine their spread in biologically relevant settings. In fact, understanding the evolutionary and ecological consequences of a drive release may represent the major challenge for the emerging technology. As seen here, our understanding of ancient naturally occurring drivers is still extremely rudimentary despite decades of research[2,52]. Natural systems such as the *t* haplotype studied here may differ from human designed constructs with respect to their mode of action, effect on their host, or past coevolution with the host genome. Moreover, currently active natural drivers may be unusual because they, by definition, only comprise of systems that have not (yet) gone extinct or to fixation, in some cases for millions of years. In contrast, the intended use of synthetic drive typically spans short ecological time scales. Despite these caveats, we think that the study of natural drive systems can provide important practical insight. Our work directly demonstrates how behavioural and population level processes can play a key role in how a drive construct spreads under real-world conditions. More specifically, we think our work holds three lessons for synthetic drive development. First and most directly, polyandry will need to be considered whenever the drive mechanism directly targets spermatogenesis, such as sex-chromosome shredders[53,54], certain toxin-antidote systems[3], or segregation distorters[55]. In house mice, the insertion of the mammalian male sex-determination gene *Sry* on the *t* haplotype has been suggested as a means to control invasive populations by turning them into males[56,57]. House mice are considered a major pest species whose global expansion is particularly damaging to endemic fauna on islands[58]. Our results suggest that polyandry may thwart the spread of such a *t-Sry* construct substantially, even to a point that may render entire release campaigns ineffective. Second and more broadly, even if the drive mechanism does not affect sperm directly, we think that polyandry is relevant for the many drive constructs that aim to suppress pests by imposing a reproductive cost to its (male) carriers (reduced fertility, sterility, or sex ratio bias[2,5]). In target species with high polyandry levels, females may simply bypass fertility losses by mating with multiple males, thereby increasing the chances of copulation with unaffected wildtype males[59]. Third and perhaps most disconcertingly, we have seen here that the release of a driver could trigger an evolutionary response in the target species, potentially altering characteristics as fundamental as the mating system. Understanding the impact of gene drive in natural populations not only provides fascinating insight into the evolutionary process, but also help us identify the factors that may make them effective as control tools.

## Methods

**Study population and data collection**. The data were collected in a free-living population of house mice inhabiting a 72 m² farm building near Zurich (see König and Lindholm[60] for a detailed description). The population was founded in 2002 by 12 individuals caught from the surrounding area and has been intensively monitored ever since. Mice are provided with nesting opportunities (40 artificial nest boxes), nesting material and *ad libitum* food and water. Vertical metal plates with passage holes, bricks, plastic tubes, and branches structure the environment and provide additional hiding places. Small openings in the walls and roof allow mice to freely leave and enter the population. None of the avian and mammalian predators are able to access the building, but predators are regularly observed in close vicinity. The population set-up is thought to closely resemble the natural habitat of house mice as they typically live commensally with humans, thus in places where food and nesting opportunities are available in abundance[61].

**Monitoring reproduction**. Reproductive activity of the mice has been closely monitored since the population was set up in 2002. Nest boxes are checked for newly born litters on a weekly basis. (Re-)capturing of mice at subadult and adult stage (see below) suggest that we detect more than 95% of pups born in the population using this method. Newly detected litters are documented and age determined based on morphological characteristics[40,60]. At 13 days of age, before pups begin to be mobile, tissue samples are collected from every pup that survived until that stage for subsequent genetic analysis (see below). About every 7 weeks, the entire population is captured, sexed, and individually marked, allowing us to estimate the overall density in the population. For the purpose of this study, we have focused on 3127 pups born in 1015 litters from 279 females that were born in the 4.5 year period between January 2006 and June 2010.

**Genetic analyses**. Parentage of all sampled pups was assigned using 25 polymorphic microsatellite markers distributed across the mouse genome (see Auclair et al.[20] for marker and PCR details). Parentage analyses were performed using Cervus 3.0[62]. We assembled candidate mother lists for each offspring based on those females that were present within two days of the offspring's estimated birthdate. Candidate father lists included all males present at the estimated time of conception. As the gestation period in mice is typically 19 days[61] but is extended following postpartum fertilisation[63], we defined the time of conception as 17–26 days before birth. Parentage assignments were only accepted at a 95% level of confidence and only when no more than one mismatching allele occurred between parent and offspring. Based on parentage assignment, a near complete pedigree is available for the entire population. The *t* genotype of an individual was identified on the basis of a microsatellite marker (*Hba-ps4*) that contains a *t* haplotype specific 13 base-pair insertion[64].

**Measuring polyandry**. Parentage information allowed us to identify litters that were sired by more than one male. Although the number of fathers per litter varied between 1 and 4 (Fig. 1), we treated a female's polyandrous tendency as a binary response trait for the purposes of this study. Accordingly, each litter was categorised either as (genetically) monandrous if sired by one male or (genetically) polyandrous if sired by more than one male. Litters of size one were excluded from the analysis as they cannot have multiple sires. Note that inference of polyandry via paternity is likely an underestimate of the actual (behavioural) polyandry rate since not every male a female mates with will succeed in fertilisation. To remind us of this important discrepancy, we refer to our measure as genetic polyandry.

The occurrence of genetic polyandry was modelled as a function of several genetic and environmental factors in a generalised animal model using a logit link and a binomial error distribution (see Supplementary Note 1 for more details). Briefly, generalised animal models are a specific type of a generalised linear mixed effects model (GLMM) that uses a pairwise relatedness matrix (derived from the pedigree) as a random effect variable[65]. This allowed us to specifically estimate the additive genetic variance $V_A$ of genetic polyandry. Moreover, as several females reproduced more than once during the observation period, we could also estimate individual differences in genetic polyandry rates between females by fitting maternal identity as a second random effect variable. The phenotypic variance explained by maternal identity is usually termed $V_{PE}$ (for permanent environment). Additional to the two random terms, we investigated the effect of female *t* genotype (+/+ and +/t), adult population size (Supplementary Fig. 3) and average monthly temperature at the time when the litter was born, as well as the size of the litter at sampling (without interactions) as fixed explanatory variables. Note that the inclusion of litter size as an explanatory variable is critically important here, as we expect a higher detection probability of genetic polyandry in larger litters due to a reduced sampling error when litters are large. Finally, the partitioning of the overall phenotypic variance of the trait, $V_P$, into additive genetic variance $V_A$, individual variance $V_{PE}$, and the unexplained residual variance $V_R$ enabled us to calculate the heritability $h^2$ of genetic polyandry in our population (Supplementary Note 1).

Fitting an animal model for non-normally distributed traits can be challenging or impossible using conventional (restricted) maximum-likelihood methods. We thus analysed our model in a Bayesian framework, using the Markov chain Monte Carlo algorithm as implemented in the R package MCMCglmm (Hadfield et al.[66], see Supplementary Note 1 for implementation details).

**The impact of polyandry on male fitness**. We calculated the reproductive success of each male as the total number of offspring sired during the observation period which survived until 13 days of age (the time of genetic sampling, see above). Although some males overlapped with the observation period (i.e. right or left-censored), average life span of 192 days[7] is relatively short relative to the observation period. We thus consider this measure a good approximation of lifetime reproduction. To examine how sperm competition affected a male's reproductive success, we calculated intensity of sperm competition experienced by a given male, which we defined as the average number of rival sperm competitors a male encountered per reproductive event. A sperm competitor is a rival male that sired at least one offspring in the same litter as the focal male. For example, a value of 2.5 would mean that, over all reproductive events, a given male shared paternity with an average of 2.5 other males per litter. Note that this measure will, again, miss any mating that did not result in successful fertilisation. We modelled reproductive success as a function of sperm competition intensity, the male's *t* genotype, total number of reproductive events (litters with at least one offspring sired), and their

two-way interactions using a GLM assuming a quasi-Poisson distribution and an exponential link function. We performed a systematic model selection using the dredge function based on qAIC-values.

**The impact of polyandry on drive frequency dynamics**. To assess the impact of polyandry on the frequency dynamics in the population, we measured drive frequency among pups during the observation period. Observed drive frequencies were then compared against theoretical frequency predictions based on monandry or polyandry. The theoretical model, which is described in detail in Supplementary Note 3, is based on the previous modelling of the system[6,7,67]. In particular, this constitutes an update on Manser et al.[7], where we have reported a rapid decline in $t$ frequency in the study population up until June 2008, and argued that sperm competition is a likely explanation for this frequency drop. Here, we measured the frequency dynamics for an additional 2 years (up to June 2010). Moreover, and unlike in the previous study, we now have solid estimates for both male sperm competitiveness (as measured in controlled lab experiments, Sutter and Lindholm[11]) and polyandry rates (as measured in this study), allowing us to make $t$ frequency predictions based on a fully parameterised model (see Supplementary Note 4 for details on parameter estimation). Note that both the monandry and polyandry model assume random mating, as we do not have empirical evidence that females show a precopulatory preference with respect to $t$ genotype in our population[49,50].

**The impact of polyandry on female fitness**. To measure whether the sperm competitive effect described here triggered an evolutionary response in female polyandry rates, we measured both selection and heritability of (genetic) polyandry (the two key requirements for evolution of a trait). To measure selection, we modelled a female's (genetic) polyandry rate (as measured by our first analysis) as a function of reproductive success (Supplementary Note 2). Heritability of polyandry was measured using the animal model (described in 'Measuring Polyandry').

**Software**. Parentage analyses were performed using Cervus 3.0. Statistical analyses were conducted in R 3.5.2 using packages MCMCglmm (v2.29), Pedantics (v1.7), MuMIn (v1.43.15), and RColorBrewer (v1.1-2). Analytical model calculations in Supplementary Note 3–4 were assisted by Maple Software 2016.

**Ethical note**. Data collection and protocols for population monitoring were approved by the Veterinary Office, Zurich, Switzerland (licences: 210/2003, 215/2006, 51/2010).

**Reporting summary**. Further information on research design is available in the Nature Research Reporting Summary linked to this article.

## Data availability
The data used in this study have been deposited on Figshare (https://doi.org/10.6084/m9.figshare.12967220). Source data are provided with this paper.

## Code availability
The scripts used for statistical analysis have been deposited on Figshare (https://doi.org/10.6084/m9.figshare.12967220).

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

## Acknowledgements
We thank the many people who have contributed to data collection on the study population. Jari Garbely has performed the laboratory work (DNA extraction and PCR). Corinne Ackermann conducted the parentage analysis. We thank Erik Postma for advice related to the animal model, and Tom Price for feedback on a previous version of this paper. We thank our funding bodies for the support—A.M.: Swiss National Science Foundation (SNF) grants P2ZHP3_161970, P300PA_177830, CRSK-3_190749, 310030M_138389, Forschungskredit of the University of Zurich, and Claraz Stiftung; B. K.: SNF grant 31003A_176114, and UZH Stiftung für wissenschaftliche Forschung; A.K. L.: SNF grants 31003A_120444, 310030M_138389, Julius-Klaus and Promotor Stiftung.

## Author contributions
A.M. and A.K.L. conceived of the study. B.K. has established and managed the longterm study with A.K.L. managing the genetic data set. A.M. performed modelling and data analyses and produced figures and text. All authors discussed the results and contributed to the paper.

## Competing interests
The authors declare no competing interests.
