## [Peer Review File · Nature Communications]

Reviewer #1 (Remarks to the Author):

This is an excellent study that provides estimates of multiple-paternity and sperm competition frequency in a wild population of house mice. The study provides data from 682 litters born to 225 females and sired by 249 males. This is an impressive data set. Multiple paternity was detected in 47% litters and the frequency of sperm competition was estimated at 60%. Mice were genotyped for whether they were carrying the t-haplotype (a meiotic drive chromosome). +/+ and +/t males had similar success in single-paternity litters but +/t males were significantly disadvantaged in multiple-paternity litters. This disadvantage was sufficient to explain the declining frequency of the t-haplotype in this population, and more generally the low frequency of t-haplotypes in most populations.

This study shows that multiple mating by females has a significant effect on the dynamics of a driving chromosome and resolves the 't paradox' that t-haplotypes are consistently found at lower frequencies than predicted by population genetic models (that assumed single-mating). This resolves a problem that has been around since at least the 1950s.

Reviewer #2 (Remarks to the Author):

This is an interesting paper which adds an important element to our understanding of naturally occurring gene drives. Although the reduction in male sperm in the case of the t-haplotype has been known for a while, its effect on sperm competition within polyandrous females in the wild has not been shown. This paper convincingly demonstrates that there is impaired sperm competition of t-allele bearing males in polyandrous females.

I am uncertain about the estimates of the intensity of sperm competition in the wild population given the known mating preferences of +/t heterozygous females favoring +/+ males. Unless there is knowledge of the parents, the inference of sperm competitive ability from data that includes single-sire broods is somewhat biased. Moreover, it is known that +-sperm from a +/t male have impaired flagellar function. Thus, even the +-sperm from +/t males is biased against fertilizing. Some type of adjustment for this bias too must be made in inferring the degree of sperm competition between genotypic males separate from t vs + allele bearing sperm.

This female mating preference for +/+ males reduces the likelihood of sperm competition between +/t males and +/+ males because it inflates the numbers of +/+ males as polyandrous partners. Since mating preferences for +/+ males over +/t males occur pre-copulation, it is a more effective method for females to avoid fertilization by t-bearing sperm than the evolution of polyandry.

The relevant number of mates per male in regard to evolutionary theory for sperm competition is the Harmonic Mean number of mates per female. The statement that, on average, the sperm of a male is competing with 0.76 other males appears based on the data in the first figure and is an arithmetic mean. The Harmonic mean is lower and sperm competition is concomitantly reduced. The data are great, but the most useful parameter for theoretical purposes to estimate from them is missing.

The model prediction made in the paper that the t-allele should die out stands in contrast to molecular genetic estimates that the t-haplotype is an ancient genetic object that arose over 1mya and is found in several subspecies of *M. musculus*. What is missing from the model that it under-estimates the life of the t-haplotype?

Synthetic gene drives do not accomplish drive in the same way that drive affects the t-haplotype in the sperm of +/t male mice. +/t males produce 95% t-bearing sperm by the killing of +-bearing sperm, reducing total sperm production relative to +/+ males. In CRISPR cas/9 synthetic drives, CR/+ heterozygotes are converted by the drive into C/C homozygotes, a much more efficient process which has no consequences for the amount of sperm C/+ males produce unless the synthetic drive targets

male fertility itself; in that case, it would target ALL sperm, not just 1/2 of them. So, I do not see the relevance to synthetic gene drives.

Reviewer #3 (Remarks to the Author):

The t-locus is arguably the most well known example of gene drive in a vertebrate. Efforts to understand why it typically occurs at frequencies well below expectation given its non Mendelian inheritance have a long and notable history with surprising little resolution. In this paper Manser et al provide convincing evidence that the t-haplotype fails to increase, and in fact decreases, in frequency due to multiple mating by females, i.e. polyandry, which occurs more frequently when population density is high. Overall, I think this is a persuasive study that provides valuable information on this classic story that has relevance, as the authors note, for efforts to engineer gene drive systems for population control.

I found the paper well written and was impressed by the large data set, sophisticated analyses, and parameterized models the authors present, some of which are described in detail in the supplement. In general, I have only a few recommendations for changes that I think could improve the ms. Below, I list them by line number.

Line 83: change "driver" to "the drive gene" or "the drive haplotype"

Line 105: What about males that failed to sire any offspring? Are they more or less likely to carry t? If they are more likely to carry the t, this would decrease the frequency of t (i.e. y in S3). As far as I can tell, whether or not a male mates is not included in the polyandry model which estimates the change in t over time (Fig. 3). Or perhaps more accurately, I believe +/+ and +/t males are assumed to mate in proportion to their frequency in the offspring. It would be helpful to know if this assumption is justified.

Line 146 and title: The term "suppression" when used in the context of gene drive typically refers to genetic modification of the drive mechanism. Here, the term is used to indicate how the mating system can limit the drive gene from increasing in frequency, as long as drive sperm are at a competitive disadvantage. To avoid potential confusion, I would encourage the authors to consider another word to describe their results. Instead of "polyandry suppresses gene drive" perhaps "polyandry prevents gene drive" or "polyandry inhibits gene drive" or "polyandry precludes gene drive". To put this another way, the t haplotype is presumably driving just fine in male carriers. It doesn't increase in frequency because the t sperm are being outcompeted in females that mate with multiple males. This is a selection process not a genetic mechanism.

Lines 319-324: As I note above, the estimate of male reproductive success excludes males that fail to father any offspring. If such males are not more or less likely to be +/t than expected by chance, this omission may have little effect. However, if they are more common than expected, then the t-frequency in the offspring will be further reduced from monandrous expectations. I think this possibility needs to be mentioned and, if possible, included.

Fig. 1 A and B use the same Y axis label, but use different numbers. Proportions are given in A while percentages are given in B. I suggest using the same numbering scheme for both.

Line 84 in Supplemental material: It's not obvious to me why litters containing less than two pups (i.e. 1 pup) were not included. While polyandry cannot be detected in such litters, they would contribute to a female's reproductive success. Given Fig S2, I doubt including such litters will alter the overall conclusions, but it seems inappropriate to me to exclude such litters, if they exist.

Reviewer Comments

Reviewer #1 (Remarks to the Author):

This is an excellent study that provides estimates of multiple-paternity and sperm competition frequency in a wild population of house mice. The study provides data from 682 litters born to 225 females and sired by 249 males. This is an impressive data set. Multiple paternity was detected in 47% litters and the frequency of sperm competition was estimated at 60%. Mice were genotyped for whether they were carrying the t-haplotype (a meiotic drive chromosome). $+/+$ and $+/t$ males had similar success in single-paternity litters but $+/t$ males were significantly disadvantaged in multiple-paternity litters. This disadvantage was sufficient to explain the declining frequency of the t-haplotype in this population, and more generally the low frequency of t-haplotypes in most populations.

This study shows that multiple mating by females has a significant effect on the dynamics of a driving chromosome and resolves the 't paradox' that t-haplotypes are consistently found at lower frequencies than predicted by population genetic models (that assumed single-mating). This resolves a problem that has been around since at least the 1950s.

Response 1 (R1). Many thanks for the positive feedback!

Reviewer #2 (Remarks to the Author):

This is an interesting paper which adds an important element to our understanding of naturally occurring gene drives. Although the reduction in male sperm in the case of the t-haplotype has been known for a while, its effect on sperm competition within polyandrous females in the wild has not been shown. This paper convincingly demonstrates that there is impaired sperm competition of t-allele bearing males in polyandrous females.

R2. Thank you for the positive and constructive feedback!

I am uncertain about the estimates of the intensity of sperm competition in the wild population given the known mating preferences of $+/t$ heterozygous females favoring $+/+$ males. Unless there is knowledge of the parents, the inference of sperm competitive ability from data that includes single-sire broods is somewhat biased.

R3. Thanks for pointing this out, mate choice is indeed a factor that has frequently been discussed in the context of the *t* haplotype (Lenington 1991). We also agree that, since we do not have *direct* information on mating success, we cannot *categorically* exclude the possibility of mate choice having an impact here, and it hence needs to be discussed (which we now do, see modifications below). However, for the following reasons, we think mate choice A) plays a minor role in our population, or, at the very least, that B) it has negligible impact on the research question addressed (impact of drive on sperm competition).

A) Is there generally evidence for +/t male avoidance in our population?

A1) No evidence from previous laboratory studies. Because of the previous reports (see Lenington 1991 for a review), we have attempted to measure mate choice under a whole series of laboratory settings over the years (published and unpublished). Despite these efforts, we have not been able to find any evidence that females from our population are distinguishing between +/t and +/+ male genotypes, nor that females adjust their remating propensity based on male *t* genotype (Sutter and Lindholm 2016, Manser et al. 2015). We currently do not know what explains the difference between our mice from the American population where olfactory preference has been reported repeatedly (Lenington 1991).

A2) No (indirect) evidence for mate choice in the data presented here. As mentioned, mate choice is difficult to measure in the paternity data used here as they are a composite measure of both mating and fertilization success. However, we do believe that the two processes do make *contrasting* predictions in analysis / figure 2, where we investigate the impact of sperm competition on male fitness (Analysis / Figure 2).

- *Mate choice prediction:* Even though we only look at a subset of males that have successfully reproduced (thus mated) at least once, we think that +/t male avoidance should result in a *decrease* in mating success in *all* males (successful and unsuccessful), thus also in the subset of males analysed here. More specifically, if females avoid +/t males prior to mating, we predict reproductive success of +/t males to be lower than that of +/+ males *independent* of the level of sperm competition. In analysis 2, this would manifest in a main effect / intercept difference between the two genotypes (+/t males have lower fitness *irrespective* of sperm competition intensity).
- *Sperm competition prediction:* In contrast, if male fitness is mainly determined by sperm competition, we expect no intercept difference (equal offspring number for single sires), but an interaction / slope difference in how sperm competition levels affect reproduction.

Analysis 2 supports the sperm competition prediction: We find *no* fitness differences in single sired litters (without sperm competition), but a stronger decline in fitness with increasing sperm competition levels in +/t males. If anything, offspring numbers for +/t among single sires are slightly elevated (although not statistically relevant).

B) More specifically, does mate choice affect the research question addressed here (how sperm competition impacts on male fitness)?

B1) There is a possibility that females of our population avoid +/t males under field conditions, but for some reason we are not able to detect and replicate this effect in our laboratory settings. However, even in this case, we think that choice would not or only marginally affect our research question (how sperm competition affects male fitness). First, we only consider males that have successfully sired offspring (and thus mated). Second, as seen above, analysis 2 can principally accommodate mate choice and sperm competition simultaneously (mate choice via intercept differences, sperm competition via slope differences). Hence, our analysis could detect a sperm competition signal (slope difference) in the presence of choice.

B2) Due to the complications of measuring +/t male sperm competitiveness under field conditions, we use the parameter estimate for sperm competitive ability from a laboratory study, where the impact of mate choice can be ruled (Sutter & Lindholm 2015).

Modifications: In light of the evidence for mate choice in the t haplotype system and the fact the two reviewers have mentioned mate choice here, we agree that it is important to critically discuss mate choice in our manuscript. At the same time, because we see little evidence for mate choice in our population, we would like to keep the general focus on the paper in terms of analysis on polyandry and sperm competition. We have made the following modifications:

1. We have added an entire section to the discussion where we critically discuss the possible impact of other factors on t frequency, with a specific focus on mate choice [LL243–265]
2. We mention in the methods section that our model is based on the assumption of random mating with respect to male t genotype [LL382–384]
3. We now mention more clearly that the estimate that goes into our model comes from lab experiments [LL131–133].

Moreover, it is known that +-sperm from a +/t male have impaired flagellar function. Thus, even the +-sperm from +/t males is biased against fertilizing. Some type of adjustment for this bias too must be made in inferring the degree of sperm competition between genotypic males separate from t vs + allele bearing sperm.

R4. This is correct, in fact, the impaired flagellar function in + sperm is the *main reason* for the low sperm competitiveness of +/t males, as rendering + sperm dysfunctional reduces the size of the functional ejaculate (almost) by half. Two scenarios are possible:

1. The t haplotype *only* impairs + sperm motility, but leaves t sperm *unaffected*. Note that *even in this scenario*, polyandry would *reduce* the probability of t fertilization. In the case of two independent monandrous matings involving a +/+ and +/t male, respectively, t sperm fertilizes disproportionately *more* eggs than expected on the quality/motility of his ejaculate alone (as the t *only* competes against impaired + sperm here). For example, if drive is 100%, t sperm fertilizes all eggs in the one monandrous mating (from the +/t male), while + sperm fertilizes all eggs in the other mating (from the +/+ male). Hence, the overall proportion of t fertilizations is 1/2. Contrast this with a polyandrous mating involving the same two males (+/+ and +/t)– now, t sperm has to compete against a *full set* of functional + from the rival male, to which they are outnumbered by 2:1. As a result, t only fertilizes 1/3 of ova. Hence, we already have a reduction in t fertilization (1/2 vs 1/3), even though t sperm are perfectly normal.
2. Sperm competition experiments between +/t and +/+ males under controlled laboratory conditions (Sutter & Lindholm (2015), Manser et al. (2017)) have shown that +/t male paternity share is significantly lower than the 1/3 predicted under scenario 1. This suggests that +/t ejaculates not only suffer the *quantitative* reduction from + sperm dysfunction, but that the t also causes an additional *qualitative* damage to its ejaculate, reducing the probability of t fertilization even further (i.e. t sperm are *also* compromised).

3. Additionally, Sutter & Lindholm (2015) did not find a *difference* in drive strength depending on whether the mating was monandrous or polyandrous. That is, *among* the offspring sired by the +/t male, the *t* was invariably passed on to ~90% (note that Manser et al. 2017 only looked at polyandrous crosses and thus could not address this question). Hence, sperm competition *between* males does not appear to affect drive levels (sperm competition *within* a male), but it does damage +/t *genotype* competitiveness between males.

Our model. Our model can accommodate both scenarios, thus *already* takes potential damage to *t* sperm into account. Sperm competitiveness parameter *r* measures the competitiveness of a +/t male *as a whole* (hence both *t* and + sperm are affected). If sperm competitiveness of the +/t male was solely dependent on the numeric loss of + sperm (scenario 1), sperm competitiveness can be directly calculated as a function of drive strength *d* directly ($r = \frac{1}{2d}$, which is 1/2 if *d*=1 in the example above). If there is additional damage to both sperm, *r* will be smaller than that (as suggested by measures). As observed empirically, our model further assumes that drive strength *within* males is independent of the mating context (monandry or polyandry).

Also note that the current paper does not aim to measure +/t male sperm disadvantage *directly*, as there are too many confounding variables without information on mating success. Rather, we here look at the overall repercussions of sperm competition for male fitness. Where our manuscript requires an exact measure of +/t male sperm competitiveness (model predictions), we use the estimates of the same population from the lab study (Sutter & Lindholm, 2015).

Modifications: We made the following modifications to the main text and supplementary material which we hope will clarify the issue.

1. We added a sentence in the introduction which mentions that +/t ejaculates perform more poorly than expected by + sperm incapacitation alone [LL73-75].
2. We elaborate the sperm competition estimate that has gone into the model prediction where we describe the model results, and highlight more clearly that it is based on measures from a different study [LL129-137].
3. In the supplementary material, we explain the quality / quantity distinction (scenario 1 and 2 above) based on the Sutter & Lindholm (2015) study [SI:LL203-209].

This female mating preference for +/+ males reduces the likelihood of sperm competition between +/t males and +/+ males because it inflates the numbers of +/+ males as polyandrous partners. Since mating preferences for +/+ males over +/t males occur pre-copulation, it is a more effective method for females to avoid fertilization by *t*-bearing sperm than the evolution of polyandry.

R5. See R2 for a general response to the mate choice point. Moreover, we do not agree that mate choice is *necessarily* the more effective strategy for female *t* avoidance.

1. Mate choice requires a *phenotypic* signal that *reliably* indicates the *t* genotype of a given male. It is not clear whether a) such a signal exists in the *t* haplotype system and b) which process would

maintain the *honesty/reliability* of the signal under directional sexual selection (Manser et al. 2017).

2. Mating decisions may not always be under complete female control- depending on the male dominance hierarchy / territory situation, a female may be forced to mate with the dominant male in her territory, irrespective of his *t* genotype, for example to avoid infanticide. In such circumstances, polyandry may provide a viable option to avoid *t* fertilisation even when matings are forced / constrained (see Auclair et al. (2014) for an example).
3. Polyandry is a powerful process by itself, that, importantly, does not require females to be able to tell apart male *t* genotypes, as low sperm competitiveness is a *pleiotropic* effect of the *t* haplotype. *+/t* carrying males only manage to fertilise ~11% of offspring when competing against *+/+* males, which is comparable to very strong precopulatory biases reported in the literature.
4. Most importantly, we have no laboratory evidence that females distinguish or are choosy with respect to *t* genotype in our study populations (see R2).

Modifications: See R2.

The relevant number of mates per male in regard to evolutionary theory for sperm competition is the Harmonic Mean number of mates per female. The statement that, on average, the sperm of a male is competing with 0.76 other males appears based on the data in the first figure and is an arithmetic mean. The Harmonic mean is lower and sperm competition is concomitantly reduced. The data are great, but the most useful parameter for theoretical purposes to estimate from them is missing.

R6. Thank you for the suggestion. Alongside the arithmetic mean, we now also report the harmonic mean of the number of sperm competitors a male encountered [LL109–112].

The model prediction made in the paper that the *t*-allele should die out stands in contrast to molecular genetic estimates that the *t*-haplotype is an ancient genetic object that arose over 1mya and is found in several subspecies of *M. musculus*. What is missing from the model that it under-estimates the life of the *t*-haplotype?

R7. We believe that our *general* polyandry model is compatible with a) the observed extant frequency measures and b) the old age of *t* haplotypes. It is important to distinguish between the general polyandry model predictions and the specific predictions for the study population here.

Specific predictions for study population: As a result of the relatively high population density, polyandry rates in our study population were unusually high for house mice (as mentioned in LL148--56), and, consequently, *t* extinction was predicted *here*, but this will be a relatively rare scenario across natural populations.

General polyandry model predictions: The general polyandry model (see Figure 3B) predicts *t* extinction only for a relatively narrow 'corner' (top left) of the parameter space. That is, under scenarios where polyandry rates are relatively high and *+/t* male sperm competitiveness sufficiently low. In the

supplementary (equation S9), we calculate t invasion thresholds more formally, and show that t goes extinct when polyandry rate (π)

$$\pi > \frac{d - \frac{1}{2}}{d} \frac{1 + r}{1 - r}$$

We have robust lab estimates for both parameters: drive strength $d=0.912$ and sperm competitiveness $r=0.126$ (Text S4, Sutter & Lindholm 2015). Inserting these estimates, we get a t extinction threshold at a polyandry rate (π) of 58%. Based on other reported empirical estimates, polyandry rates are likely to fall *below* this threshold in *most* mouse populations. In these scenarios (polyandry 0-57%), polyandry will *reduce* t frequency, but not push it to extinction, which is very much in line with extant frequency observations from natural populations and the estimated age of the t .

Modifications: We have made the following modifications that hopefully clarify this point.

1. We explicitly mention the eradication threshold in the model results [LL129–137]
2. We again mention the extinction threshold where we talk about the t frequency paradox in the discussion [LL237–239].

Synthetic gene drives do not accomplish drive in the same way that drive affects the t -haplotype in the sperm of $+/t$ male mice. $+/t$ males produce 95% t -bearing sperm by the killing of $+/+$ -bearing sperm, reducing total sperm production relative to $+/+$ males. In CRISPR $cas/9$ synthetic drives, $CR/+$ heterozygotes are converted by the drive into C/C homozygotes, a much more efficient process which has no consequences for the amount of sperm $C/+$ males produce unless the synthetic drive targets male fertility itself; in that case, it would target ALL sperm, not just 1/2 of them. So, I do not see the relevance to synthetic gene drives.

R8. We agree with the reviewer that polyandry may not be important for *all* classes of synthetic drive constructs, in particular homing based CRISPR drives with cargo that have no / or little fertility effects (replacement drives). However, this does not mean that our work is irrelevant for all constructs currently discussed or developed. There is currently strong research interest for alternatives to the described CRISPR homing drives, which appear extremely vulnerable to rapid resistance formation via non-homologous end-joining. Overall, we see two specific areas where our work can provide important insight for synthetic drive research:

1. A number of the currently discussed drive synthetic mechanisms *do* target gamete production and spermatogenesis. The t -*Sry* discussed in house mice is an obvious example, for which our work is directly relevant [LL279–281]. X or Y shredders that have, for example, been developed in the malaria vector *Anopheles gambiae* also target X-chromosomes during spermatogenesis (Galizi et al. 2014; 2016). While the authors assert that their constructs have little effect on fertility, it seems unclear as to what would happen if carriers are exposed to sperm competition. Indeed, there is some evidence that malaria vectors are polyandrous, although a different genus (*Aedes*, Helinski et al., 2012). Just this year, Champer et al. (2020) have proposed a number of

toxin-antidote systems, among which is a mechanism that targets expression critical for successful spermatogenesis (termed Toxin-Antidote Dominant Sperm, TADS).

2. More generally, even if the drive mechanism *itself* does not damage gamete production, we think our work is still relevant in the context of suppression gene drives that target host reproduction via a fertility cost in the cargo (e.g. sex ratio bias, reduced fertility, sterility). In these cases, the mating system of a target species (of which polyandry is a crucial aspect) will have obvious repercussions on the effectiveness of a drive intervention, such that species where low fertility individuals are easily bypassed via repeated mating may simply be less vulnerable to such drive interventions (Prowse 2019).

Also note that Reviewer 3 has explicitly acknowledged the relevance of our work for applied drive research.

Modifications. We have modified the last discussion paragraph along those lines, hopefully this clarifies the relevance of our work for synthetic drive design [LL273–285].

Reviewer #3 (Remarks to the Author):

The t-locus is arguably the most well known example of gene drive in a vertebrate. Efforts to understand why it typically occurs at frequencies well below expectation given its non Mendelian inheritance have a long and notable history with surprising little resolution. In this paper Manser et al provide convincing evidence that the t-haplotype fails to increase, and in fact decreases, in frequency due to multiple mating by females, i.e. polyandry, which occurs more frequently when population density is high. Overall, I think this is a persuasive study that provides valuable information on this classic story that has relevance, as the authors note, for efforts to engineer gene drive systems for population control.

I found the paper well written and was impressed by the large data set, sophisticated analyses, and parameterized models the authors present, some of which are described in detail in the supplement. In general, I have only a few recommendations for changes that I think could improve the ms. Below, I list them by line number.

R9. Many thanks for the positive and constructive feedback!

Line 83: change “driver” to “the drive gene” or “the drive haplotype”

R10. Changed [L86].

Line 105: What about males that failed to sire any offspring? Are they more or less likely to carry t? If they are more likely to carry the t, this would decrease the frequency of t (i.e. y in S3). As far as I can tell, whether or not a male mates is not included in the polyandry model which estimates the change in t over time (Fig. 3). Or perhaps more accurately, I believe +/+ and +/t males are assumed to mate in proportion to their frequency in the offspring. It would be helpful to know if this assumption is justified.

R11. There is one earlier study on our study population (Lindholm et al. 2013) that has addressed this very question, and it has indeed found that +/t males are indeed less likely to mate compared to +/+ males. However, the fundamental problem here is that, since mating success was estimated by paternity outcomes (and hence pre- and postmating processes are conflated), it is really difficult to disentangle whether this bias is a result of actual mate choice or systematically lower fertilization probability given successful matings (unsuccessful males may mate, but not sire offspring). Directly observing matings in the wild population is nearly impossible, but we have since measured mate choice in various settings under laboratory conditions, none of which have found evidence that females of our population seem to be able to distinguish between the two male genotypes. On the whole, and contrary reports from other mouse population (Lenington 1991), we thus think there is no evidence that mate choice plays a role in our population. For this reason, as already mentioned under R2, we want to focus on sperm competition in terms of analysis.

Modifications:

1. We now mention explicitly in the main text that our model assumes random mating [LL282–284]
2. We dedicate an entire paragraph in the discussion other potential factors that may impact *t* frequency, with a focus on mate choice and the findings by Lindholm et al. (2013) [LL243–265]

Line 146 and title: The term “suppression” when used in the context of gene drive typically refers to genetic modification of the drive mechanism. Here, the term is used to indicate how the mating system can limit the drive gene from increasing in frequency, as long as drive sperm are at a competitive disadvantage. To avoid potential confusion, I would encourage the authors to consider another word to describe their results. Instead of “polyandry suppresses gene drive” perhaps “polyandry prevents gene drive” or “polyandry inhibits gene drive” or “polyandry precludes gene drive”. To put this another way, the *t* haplotype is presumably driving just fine in male carriers. It doesn’t increase in frequency because the *t* sperm are being outcompeted in females that mate with multiple males. This is a selection process not a genetic mechanism.

R12. Point taken, this is an issue that we have discussed previously when thinking about a title, and went with the term suppression because we were not very happy with the alternatives we could think of. However, we agree with the reviewer that the term is misleading, and we have now changed the title to clarify that we do not mean molecular/genetic suppression of the driver [LL1–2].

Lines 319-324: As I note above, the estimate of male reproductive success excludes males that fail to father any offspring. If such males are not more or less likely to be +/t than expected by chance, this omission may have little effect. However, if they are more common than expected, then the *t*-frequency in the offspring will be further reduced from monandrous expectations. I think this possibility needs to be mentioned and, if possible, included.

R13. We agree that a female mating preference for +/+ wildtype males would further reduce *t* frequency. However, as mentioned above (R2), we do not see evidence for such a mating bias when we test animals

derived from the study population under controlled laboratory condition. We think that laboratory experiments are much more suited to measure mating preferences as, unlike in the wild population, mating (behavior) can be observed directly, and do not have to be inferred indirectly via paternity analysis. Furthermore (and as a consequence of little empirical evidence), we would like to keep the main focus of our manuscript on the effect of polyandry and sperm competition. A whole host of factors have been proposed to explain the t frequency paradox (inbreeding, dispersal, survival effects, metapopulation structure, male dominance). We see the main message of our manuscript in demonstrating that polyandry is likely to play a key (but not necessarily exclusive) role in explaining low t frequencies. Within this remit, we believe that only looking at males that were successful in fathering offspring is appropriate.

Modifications: As mentioned above (R2 and R11), we have added a section to the discussion where we discuss other factors that may impact on t frequency in the wild populations, with a particular focus on mate choice. We also clarify there that we do not consider polyandry to be the *only* factor impacting on t frequency [LL243–265].

Fig. 1 A and B use the same Y axis label, but use different numbers. Proportions are given in A while percentages are given in B. I suggest using the same numbering scheme for both.

R14. Thanks for spotting this. The axis label between the two panels was meant to depict absolute litter numbers for panel A. We agree that this was confusing, and have now added an additional axis on subfigure B for clarification [Fig. 1].

Line 84 in Supplemental material: It's not obvious to me why litters containing less than two pups (i.e. 1 pup) were not included. While polyandry cannot be detected in such litters, they would contribute to a female's reproductive success. Given Fig S2, I doubt including such litters will alter the overall conclusions, but it seems inappropriate to me to exclude such litters, if they exist.

R15. Thank you for the suggestion. We agree with the reviewer and have now updated our analysis [SI:LL84–88, Tab S1, and Fig S2]. As suspected, the addition of litters with one pup only did not affect the overall conclusions. Note that the overall number of females remained the same (we just include more of their litters) as we cannot derive a polyandry estimates for females who only had a single litter of size 1.

References

- Auclair, König, and Lindholm (2014). Socially mediated polyandry: a new benefit of communal nesting in mammals. *Behavioral Ecology*, 25(6):1467–1473.
- Champer, Kim, Champer., Clark, and Messer (2020). Performance analysis of novel toxin-antidote crispr gene drive systems. *BMC biology*, 18(1):1–17.
- Galizi, Doyle, Menichelli, Bernardini, Deredec, Burt, Stoddard, Windbichler, and Crisanti (2014). A synthetic sex ratio distortion system for the control of the human malaria mosquito. *Nature Communications*, 5:3977.
- Galizi, Hammond, Kyrou, Taxiarchi, Bernardini, O'Loughlin, Papathanos, Nolan, Windbichler, and Crisanti (2016). A crispr-cas9 sex-ratio distortion system for genetic control. *Scientific reports*, 6:31139.
- Helinski, M. E., Valerio, L., Facchinelli, L., Scott, T. W., Ramsey, J., and Harrington, L. C. (2012). Evidence of polyandry for *aedes aegypti* in semifield enclosures. *The American journal of tropical medicine and hygiene*, 86(4):635–641
- Lenington (1991). The t complex: a story of genes, behavior, and populations. *Advances in the Study of Behaviour*, 20:51–86.
- Manser, König, and Lindholm (2015). Female house mice avoid fertilization by t haplotype incompatible males in a mate choice experiment. *Journal of Evolutionary Biology*, 28(1):54– 64.
- Manser, Lindholm, Simmons, and Firman (2017). Sperm competition suppresses gene drive among experimentally evolving populations of house mice. *Molecular Ecology*, 14:189.
- Prowse, T. A., Adikusuma, F., Cassey, P., Thomas, P., and Ross, J. V. (2019). A y-chromosome shredding gene drive for controlling pest vertebrate populations. *Elife*, 8:e41873.
- Sutter & Lindholm (2015). Detrimental effects of an autosomal selfish genetic element on sperm competitiveness in house mice. *Proceedings of the Royal Society B*, 282(1811):20150974.
- Sutter & Lindholm (2016). No evidence for female discrimination against male house mice carrying a selfish genetic element. *Current Zoology*, 62(6):zow063

REVIEWERS' COMMENTS:

Reviewer #1 (Remarks to the Author):

I approved of the manuscript in the first round and have not changed my judgment

Reviewer #2 (Remarks to the Author):

Thank you for so thoroughly addressing my concerns.

Reviewer #4 (Remarks to the Author):

My compliments on a well crafted paper.

Jerry Wilkinson

Reviewer #1 (Remarks to the Author):

I approved of the manuscript in the first round and have not changed my judgment

Reviewer #2 (Remarks to the Author):

Thank you for so thoroughly addressing my concerns.

Reviewer #4 (Remarks to the Author):

My compliments on a well crafted paper.

Jerry Wilkinson

Response: Many thanks for the positive, encouraging, and helpful feedback on our manuscript.